# Dual interaction of scaffold protein Tim44 of mitochondrial import motor with channel-forming translocase subunit Tim23

See-Yeun Ting, Nicholas L Yan, Brenda A Schilke, Elizabeth A Craig*

Department of Biochemistry, University of Wisconsin-Madison, Madison, United States

**Abstract** Proteins destined for the mitochondrial matrix are targeted to the inner membrane Tim17/23 translocon by their presequences. Inward movement is driven by the matrix-localized, Hsp70-based motor. The scaffold Tim44, interacting with the matrix face of the translocon, recruits other motor subunits and binds incoming presequence. The basis of these interactions and their functional relationships remains unclear. Using site-specific in vivo crosslinking and genetic approaches in *Saccharomyces cerevisiae*, we found that both domains of Tim44 interact with the major matrix-exposed loop of Tim23, with the C-terminal domain (CTD) binding Tim17 as well. Results of in vitro experiments showed that the N-terminal domain (NTD) is intrinsically disordered and binds presequence near a region important for interaction with Hsp70 and Tim23. Our data suggest a model in which the CTD serves primarily to anchor Tim44 to the translocon, whereas the NTD is a dynamic arm, interacting with multiple components to drive efficient translocation.

*For correspondence: ecraig@wisc.edu

**Competing interests:** The authors declare that no competing interests exist.

## Introduction

Mitochondria are essential organelles necessary for a variety of critical metabolic functions. Since the vast majority of proteins in the mitochondria are encoded by nuclear DNA, efficient translocation of proteins from the cytoplasm is required for robust mitochondrial function (*Sickmann et al., 2003*). Proteins destined for the matrix initially enter mitochondria via the TOM complex of the outer membrane, and are then translocated across the inner membrane into the matrix through the Tim17/23 translocon (*Endo and Yamano, 2009*; *Schmidt et al., 2010*). Most matrix-bound proteins are synthesized as preproteins with a positively charged amino-terminal targeting sequence (presequence), which interacts sequentially with 'receptor' proteins in transit from the cytosol to the inner membrane translocon (*Bajaj et al., 2014*; *Lytovchenko et al., 2013*; *Marom et al., 2011b*; *Shiota et al., 2011*). Upon insertion of the presequence into the translocon, it is driven into the matrix by the membrane potential across the inner membrane. Translocation of the remainder of the preprotein requires action of the mitochondrial import motor, which resides on the matrix side of the inner membrane (*Chacinska et al., 2009*; *Marom et al., 2011a*; *Neupert, 2015*; *Schulz et al., 2015*).

The Tim17/23 translocon is composed of two essential and related integral membrane proteins. Both Tim17 and Tim23 have four membrane-spanning helices and two hydrophilic loops facing the matrix (*Alder et al., 2008*; *Bauer et al., 1996*). In addition, Tim23 has an N-terminal domain in the intermembrane space (IMS) that interacts with presequence (*de la Cruz et al., 2010*). The translocon maintains the potential across the inner membrane. Tim23 forms the protein-conducting pore itself, whereas Tim17 contributes to the structure and gating of the channel (*Bauer et al., 1996*; *Martinez-Caballero et al., 2007*; *Meier et al., 2005*; *Ryan et al., 1998*; *Truscott et al., 2001*). Upon

interaction with a presequence, the channel opens to allow passage of the incoming polypeptide, driven by the membrane potential (*Truscott et al., 2001*). The driving force of the motor for translocation of the remainder of the protein is provided by the ATP-hydrolyzing 70 kDa heat shock protein (mtHsp70) (*Gambill et al., 1993*; *Kang et al., 1990*). Like all Hsp70s, mtHsp70's innate ATPase activity is weak and stimulated by a J-protein co-chaperone. This hydrolysis drives conformational changes in mtHsp70, serving to stabilize interaction with the translocating polypeptide. Pam18 (Tim14) is the J-protein co-chaperone that serves this function for the import motor (*D'Silva et al., 2003*; *Truscott et al., 2003*). It is a transmembrane protein that interacts with Tim17 in the IMS and with Pam16 (Tim16) in the matrix (*Chacinska et al., 2005*; *Schilke et al., 2012*). Neither mtHsp70, Pam18 nor Pam16 directly interact with the translocon in the matrix. Rather, the essential peripheral inner membrane protein, Tim44, serves as a scaffold, bridging the translocon and other motor components (*Schiller et al., 2008*; *Ting et al., 2014*). The motor also contains the nonessential, transmembrane protein, Pam17, thought to be involved in early steps of the translocation process (*Hutu et al., 2008*; *Schiller, 2009*). Pam17 does not interact with Tim44, but directly with matrix-exposed loop 1 of Tim17 (*Ting et al., 2014*).

Tim44 is a two-domain protein, consisting of the N-terminal (residues 43 to 209) and C-terminal (residues 210 to 431) domains, referred to as NTD and CTD, respectively, with the first 42 residues being the cleavable presequence. Structural information about the NTD is lacking. However, evidence indicates that it interacts with Tim23, mtHsp70 and Pam16 (*Schilke et al., 2012*; *Schiller et al., 2008*). The function of the CTD is less clear. However, structural information is available (*Figure 1A*) (*Handa et al., 2007*; *Josyula et al., 2006*). The CTD is composed of both α-helices and β-strands, with the core forming an α+β barrel. Two N-terminal α-helices, suggested to be involved in membrane binding (*Marom et al., 2009*), protrude from the core. Although much progress has been made in understanding Tim44 function, large gaps remain. For example, although recent data indicates that the CTD interacts with Tim17 (*Banerjee et al., 2015*), the basis of the interaction and importance is unknown. Likewise, precursor proteins have been crosslinked to Tim44 *in organellar* (*Blom et al., 1993*; *Gruhler et al., 1997*; *Kanamori et al., 1997*; *Scherer et al., 1992*) and presequence peptides to purified Tim44 in vitro (*Marom et al., 2011b*), raising the possibility that Tim44 plays a regulatory role in transferring translocating polypeptides to mtHsp70, activating the motor. However, no information is available regarding where on Tim44 the presequence interaction occurs. Lack of such information impedes our understanding of the mechanism of this critical molecular machine. With a goal of overcoming these deficits, we utilized a variety of experimental approaches. We report three key findings regarding Tim44 function: the CTD specifically crosslinks to both Tim23 and Tim17; both the NTD and the CTD crosslink to the matrix exposed loop 1 of Tim23 in vivo; the NTD, which we found to be intrinsically disordered, but not the CTD, interacts with presequence. These new data allow presentation of a more comprehensive model of how the action of the translocon and import motor are coupled to drive import of proteins into the mitochondrial matrix.

## Results

### Residue D345 in the Tim44 CTD is important for translocon association

As a first step in better defining the roles of the Tim44 CTD, we carried out a screen for temperature sensitive (ts) Tim44 mutants with a goal of identifying functionally important surface exposed residues. Screening an error-prone PCR library of *TIM44*, we identified an aspartic acid to glycine alteration at position 345 that met these criteria. $tim44_{D345G}$ grows poorly at 37°C (*Figure 1B*). D345 is surface exposed, located in the linker region between the first long β-strand (β1a) of the CTD's antiparallel β-sheet and the adjacent, shorter β-strand (β1b). Alignment of Tim44 sequences from a wide range of eukaryotes revealed that the aspartic acid is conserved (*Figure 1C*).

We further tested the functional importance of residue 345 by substituting alanine or lysine for aspartate. $tim44_{D345A}$ had no obvious growth phenotype at 37°C. $tim44_{D345K}$ had a significant growth defect even at 30°C, and did not form colonies at 37°C. All three Tim44 variants were expressed at a level similar to that of wild type (wt) protein (*Figure 1D*). To assess whether substitutions at position 345 caused a defect in mitochondrial protein import, accumulation of the precursor form of the mitochondrial matrix protein Hsp60 after transferring of cells to 37°C was monitored

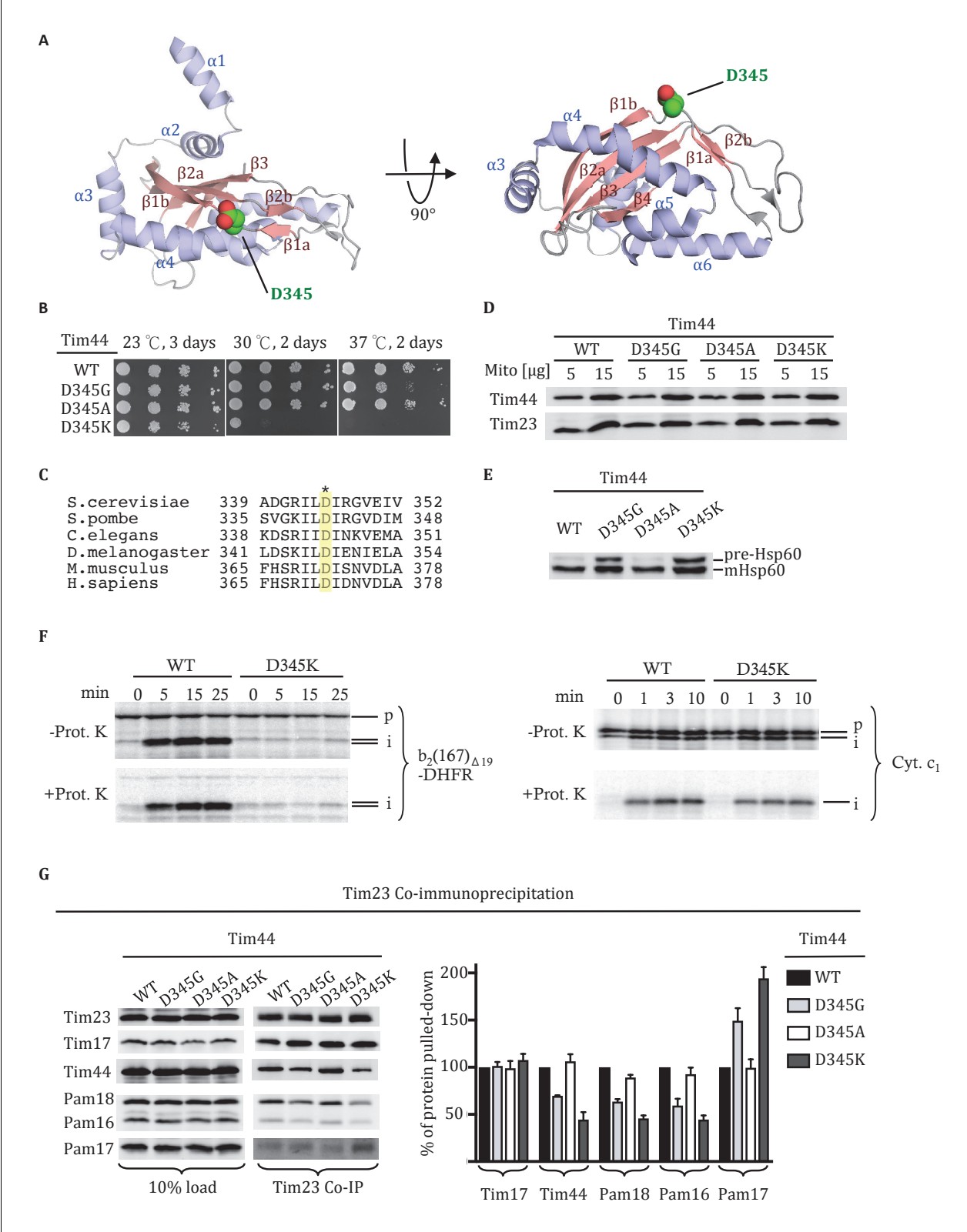

**Figure 1.** Residue D345 in the CTD of Tim44 is important for cell viability and association of the import motor with the translocon. (**A**) Structure of the CTD of Tim44; PDB entry 2FXT (*Josyula et al., 2006*). α-helices and β-strands are labeled. Residue D345 is indicated and shown in sphere representation. Right view of the structure, rotated 90° along the horizontal axis from the orientation on the left. (**B**) Growth phenotype. Ten-fold serial dilutions of *tim44-Δ* cells expressing the indicated Tim44 variants were plated on minimal media and incubated at the indicated temperatures and

*Figure 1 continued on next page*

*Figure 1 continued*

times. (C) Sequence alignment of Tim44 from indicated eukaryotic species was done using Clustal Omega; residue D345 is indicated (*). (D) 5 and 15 μg of mitochondria isolated from Tim44 mutants were analyzed by SDS–PAGE, followed by immunoblotting against Tim23-and Tim44-specific antibodies. (E) Precursor accumulation. Whole cell lysates of the indicated *TIM44* mutants were prepared from cells grown at 30°C and shifted to 37°C for 6 hr. Proteins separated by SDS–PAGE were subjected to immunoblot analysis using antibodies specifiic to Hsp60. The Hsp60 precursor form (pre-Hsp60) and mature form (mHsp60) are indicated. (F) $^{35}$S-labeled precursors of cytochrome b2-(167)$_{\Delta 19}$-DHFR and cytochrome $c_1$ were imported into wild-type (WT) and $tim44_{D345K}$ mutant mitochondria at 25°C. Where indicated, mitochondria were subsequently treated with proteinase K (Prot. K). Samples were analyzed by SDS-PAGE and autoradiography. p, precursor; I, intermediate; m, mature. (G) Co-immunoprecipitation with Tim23. Mitochondria were solubilized by treatment with digitonin. Solubilized material was subjected to immunoprecipitation using Tim23-specific antibodies crosslinked to protein A beads. Precipitates were analyzed by SDS-PAGE and immunoblotting using antibodies specific for the indicated proteins. 10% of input was used as a loading control. Signals were quantified by Image J software (RRID: SCR_003070) and plotted as percentages of the wt control reaction value. Data represent the mean ± standard deviation, with n = 3. The data of the figure can also be seen in *Figure 1—source data 1*.

The following source data is available for figure 1:

**Source data 1.** Data analysis/quantification of Tim23 co-immunoprecipitation.

using immunoblotting analysis (*Figure 1E*). Hsp60 precursor accumulation was observed in $tim44_{D345G}$ and $tim44_{D345K}$, consistent with the observed growth defects. Import of precursor proteins into mitochondria isolated from wt and $tim44_{D345K}$ cells was also tested (*Figure 1F*). Two radiolabeled precursors were used: cytochrome $b_2$-(167)$_{\Delta 19}$-DHFR, a precursor protein composed of mouse DHFR fused to the N-terminal 167 amino acids of cytochrome $b_2$ with a deletion of 19 amino acids within the inner membrane sorting signal to ensure sorting to the matrix rather than the inner membrane and cytochrome $c_1$ which is sorted to the inner membrane through the Tim23 translocon in a motor-independent manner (*Hutu et al., 2008*). While the accumulation of cleaved, protease-resistant cytochrome $c_1$ were very similar in the two types of mitochondria, $tim44_{D345K}$ was severely defective in import of cytochrome $b_2$-(167)$_{\Delta 19}$-DHFR compared to wt (*Figure 1F*).

To test whether alteration of D345 affected interaction of Tim44 and other components of the motor with the Tim17/23 translocon, we carried out co-immunoprecipitation experiments using mitochondrial lysates and affinity-purified Tim23-specific antibody. As expected, the motor components, Tim44, Pam16 and Pam18, co-precipitated with Tim23 from wt lysates. However, these motor components, particularly Tim44, were reduced in co-precipitates from mitochondrial lysates of $tim44_{D345G}$ and $tim44_{D345K}$, even though the amounts of Tim23 precipitated from the two extracts were similar. (*Figure 1G*). As previously reported, binding of Pam17 with the TIM23 complex increases when the association of Tim44 decreases (*Hutu et al., 2008*; *Schilke et al., 2012*). These results indicate that alteration of residue 345 of Tim44's CTD compromises the association of the motor components with the translocon.

## Functional importance of conserved residues near D345

We extended our analysis to other conserved, surface exposed residues near D345 on β1 strand, adjacent β2 strand or α4 helix, around which the β-sheet curves. Five residues (E295, E303, R342, E350, and R370) were individually changed to alanine (*Figure 2A*, *Figure 2—figure supplement 1*). All mutants grew normally at the temperatures tested, even at 37°C (*Figure 2A*). Since $tim44_{D345A}$ also had no obvious phenotype, we decided to construct double substitution mutants, changing the D345 codon and one of the other five to alanine (*Figure 2B*). All variants were expressed at levels similar to wt Tim44 (*Figure 2C*). Two of the double alanine mutants, $tim44_{D345A/R342A}$ and $tim44_{D345A/R370A}$, had no obvious growth defect. Two others, $tim44_{D345A/E295A}$ and $tim44_{D345A/E350A}$, were ts, growing slowly at 37°C. The growth defect of $tim44_{D345A/E303A}$ was the most extreme. It did not form colonies at 30°C and grew very slowly at 23°C (*Figure 2B*). Hsp60 precursor accumulation was observed in all double alanine mutant strains that had growth defects (i.e. $tim44_{D345A/E295A}$, $tim44_{D345A/E303A}$, $tim44_{D345A/E350A}$) (*Figure 2D*). In addition, motor components, particularly Tim44, were reduced in Tim23 co-precipitates from lysates of $tim44_{D345A/E295A}$, $tim44_{D345A/E350A}$, and $tim44_{D345A/E303A}$ mitochondria (*Figure 2E*). As $tim44_{D345A/E303A}$ had the most severe defect in association of motor components, we tested import functionality using *in organellar* import assays. Similar

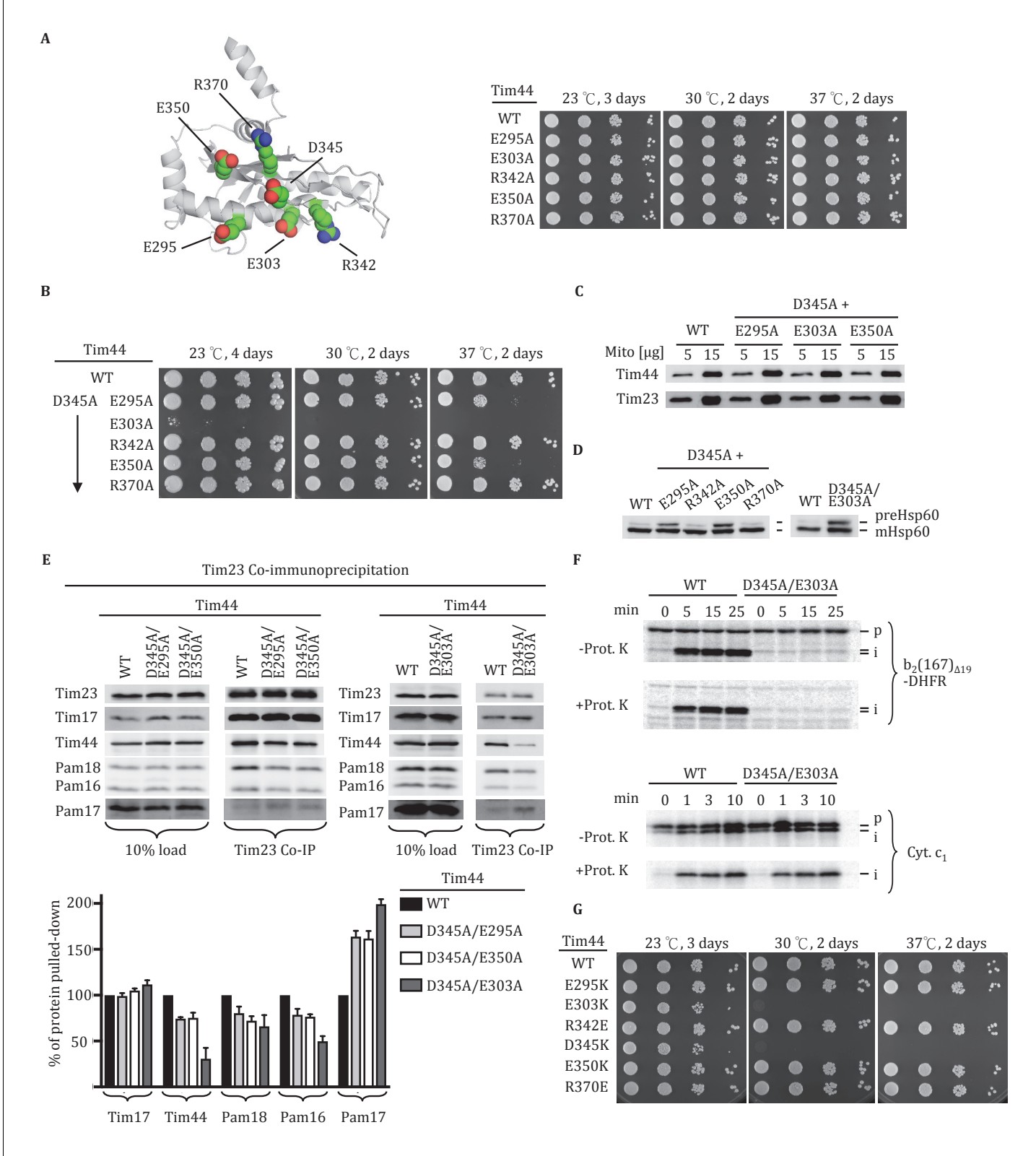

**Figure 2.** Importance of conserved residues near D345. (**A**) Structure of the CTD of Tim44; PDB entry 2FXT (*Josyula et al., 2006*). The five nearby charged residues analyzed (E295, E303, R342, E350, and R370) are indicated. (**B**) Growth phenotype. Ten-fold serial dilutions of *tim44-Δ* cells expressing the indicated Tim44 variants were plated on minimal media and incubated at the indicated temperatures and times. (**C**) 5 and 15 μg of mitochondria isolated from Tim44 mutants were analyzed by SDS–PAGE, followed by immunoblotting against antibodies to Tim23 and Tim44. (**D**) Precursor
*Figure 2 continued on next page*

*Figure 2 continued*

accumulation. Whole cell lysates from indicated cells grown at 30°C and shifted to 37°C for 6 hr (left). $tim44_{D345A/E303A}$ was grown at 23°C and then shifted to 37°C for 6 hr, because it does not grow at 30°C (right). Proteins separated by SDS–PAGE were subjected to immunoblot analysis using antibodies specific to Hsp60. The Hsp60 precursor form (pre-Hsp60) and mature form (mHsp60) are indicated. (E) Co-immunoprecipitation with Tim23. Mitochondria were solubilized by treatment with digitonin. Solubilized material was subjected to immunoprecipitation using Tim23-specific antibodies crosslinked to protein A beads. Precipitates were analyzed by SDS-PAGE and immunoblotting using antibodies specific for the indicated proteins. 10% of input for immunoprecipitation was used as a loading control. Signals were quantified by Image J (RRID: SCR_003070) software and plotted as percentages of the wild-type control reaction value. Data represent the mean ± standard deviation, with n = 3. The data of the figure can also be seen in **Figure 2—source data 1**. (F) $^{35}$S-labeled precursors of cytochrome b2-(167)$_{\Delta19}$-DHFR and cytochrome $c_1$ were imported into wild-type (WT) and $tim44_{E303A/D345A}$ mutant mitochondria at 25°C. Where indicated, mitochondria were subsequently treated with proteinase K (Prot. K). Samples were analyzed by SDS-PAGE and autoradiography. p, precursor; I, intermediate; m, mature. (G) Growth phenotype. Ten-fold serial dilutions of $tim44$-$\Delta$ cells expressing the indicated Tim44 variants were plated on minimal media and incubated at the indicated temperatures and times.

The following source data and figure supplement are available for figure 2:

**Source data 1.** Data analysis/quantification of Tim23 co-immunoprecipitation.
**Figure supplement 1.** Sequence alignment of Tim44 from indicated eukaryotic species was done using Clustal Omega.

to $tim44_{D345K}$, $tim44_{D345A/E303A}$ mitochondria were defective in import of cytochrome b2-(167)$_{\Delta19}$-DHFR (**Figure 2F**).

Finally, to better understand the relative importance of the individual residues for the function of Tim44, we constructed additional *TIM44* mutants: We individually substituted these five residues to ones of opposite charge (i.e. to lysine or glutamate). In agreement with the double alanine substitutions described above, E303K substitution resulted in the most severe growth defect of the cells, similar to the D345K substitution (**Figure 2G**). Together, these results suggest that D345 and the nearby, surface exposed charged amino acids, especially residue E303, are functionally important for Tim44 association with the translocon.

## Crosslinking of Tim44 CTD residues to Tim23 or Tim17

The functional importance of D345 and nearby residues for association of Tim44 with the translocon led us to hypothesize that the α4-β1 surface of the CTD directly interacts with the Tim17/23 translocon. To test this idea, we carried out in vivo site-specific crosslinking, focusing on the Tim44 residues we found to be important for Tim17/23-motor association (i.e. D345, E295, E303, E350). We incorporated the photo-reactive amino acid *p*-benzoyl-L-phenylalanine (Bpa) in place of the endogenous amino acid using enhanced nonsense suppression (**Krishnamurthy et al., 2011**). To covalently crosslink binding partners, cells expressing Tim44$_{Bpa}$ were exposed to UV irradiation. Tim44 and associated proteins were pulled down using the hexa-histidine tag at Tim44's N-terminus, and separated by electrophoresis. No obvious crosslinks were detected by immunoblotting when Bpa was incorporated at positions E295 and E303. However, in the case of D345Bpa, we observed a crosslinked product migrating at ~64 kDa that reacted with antibodies for Tim44 and Tim23 (**Figure 3A**). In the case of E350Bpa, we also observed a Tim44-reactive crosslinking species. But, it was detected by antibodies specific for Tim17, not those for Tim23. Taken together, these results suggested that the CTD of Tim44 interacts with both Tim23 and Tim17.

We expanded our analysis to include additional surface exposed residues near the sites of Tim23 and Tim17 crosslinking. Bpa was successfully incorporated at 19 positions (**Figure 3—figure supplement 1**). After exposure of cells to UV, strong crosslink products migrating between 58 and 64 kDa were detected for six variants using Tim44-specific antibodies. Those having Bpa at positions I374, I385, and A391 reacted with antibodies for Tim23 (**Figure 3B**). On the other hand, lysates from cells expressing Tim44 variants having Bpa at positions S271, R272 and N286 reacted with Tim17-specific antibodies (**Figure 3C**). Taken together, these results are consistent with the idea that the CTD of Tim44 interacts with both Tim23 and Tim17, via two distinct binding interfaces (i.e. patches) (**Figure 3D**, **Figure 3—figure supplement 1**).

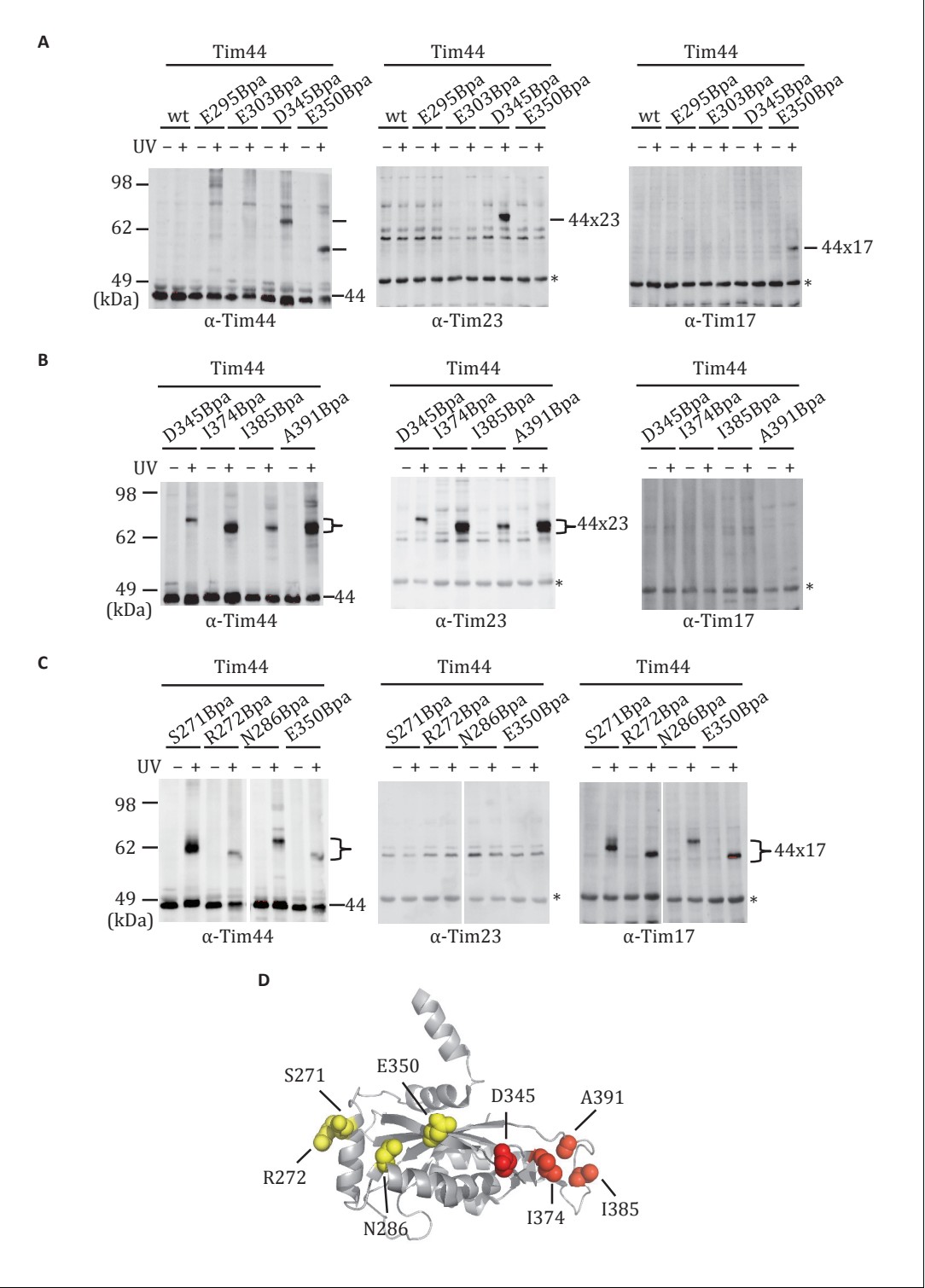

**Figure 3.** Site-specific crosslinking of Tim44 to Tim23 or Tim17. (**A,B,C**). Yeast strains expressing variants of Tim44 with Bpa incorporated at the indicated positions. After UV irradiation (+), or as a control no irradiation (-), Tim44 was affinity purified via its N-terminal $His_6$ tag, followed by SDS-PAGE and immunoblotting with the indicated antibodies. Migrations of size standard, in kDa, are indicted (left); Tim44 (44) and Tim23 (23) or Tim17 (17) reactive bands are indicated. The asterisks indicate non-specific cross-reactive bands. (**D**) The residues whose crosslinking was detected are shown in sphere representation (Red, Tim44-Tim23 crosslinks; yellow, Tim44-Tim17 crosslinks).

*Figure 3 continued on next page*

*Figure 3 continued*

The following figure supplement is available for figure 3:

**Figure supplement 1.** Summary of Tim44 CTD in vivo crosslinking.

## Both domains of Tim44 crosslink to loop 1 of Tim23

The results described above indicate that the CTD interacts with Tim23. Previous results demonstrated that the NTD is also important for association of Tim44 with the translocon (*Schiller et al., 2008*) and can be crosslinked to Tim23 using the site-specific Bpa technique (*Ting et al., 2014*). The crosslinking of both the NTD and CTD of Tim44 to Tim23 raises the question as to the sites of interaction. Tim23 has two matrix-exposed loops (*Figure 4A*), 24-residue loop 1 (residues 122 to 145) which can be crosslinked to Tim44 (*Ting et al., 2014*) and 7-residue loop 3 (residues 191 to 197). There are two obvious possibilities: (1) One Tim44 domain interacts with Tim23's loop 1, the second with loop 3; (2) both domains interact with loop 1. We were not able to directly test whether loop 3 of Tim23 can crosslink to Tim44, because translation terminated at engineered stop codons in that loop, even when Bpa was present. As an alternative, we took advantage of the previous finding that cells expressing a C-terminal Tim23 truncation lacking TM3, loop 3, and TM4 segments (*tim23Δ176-*

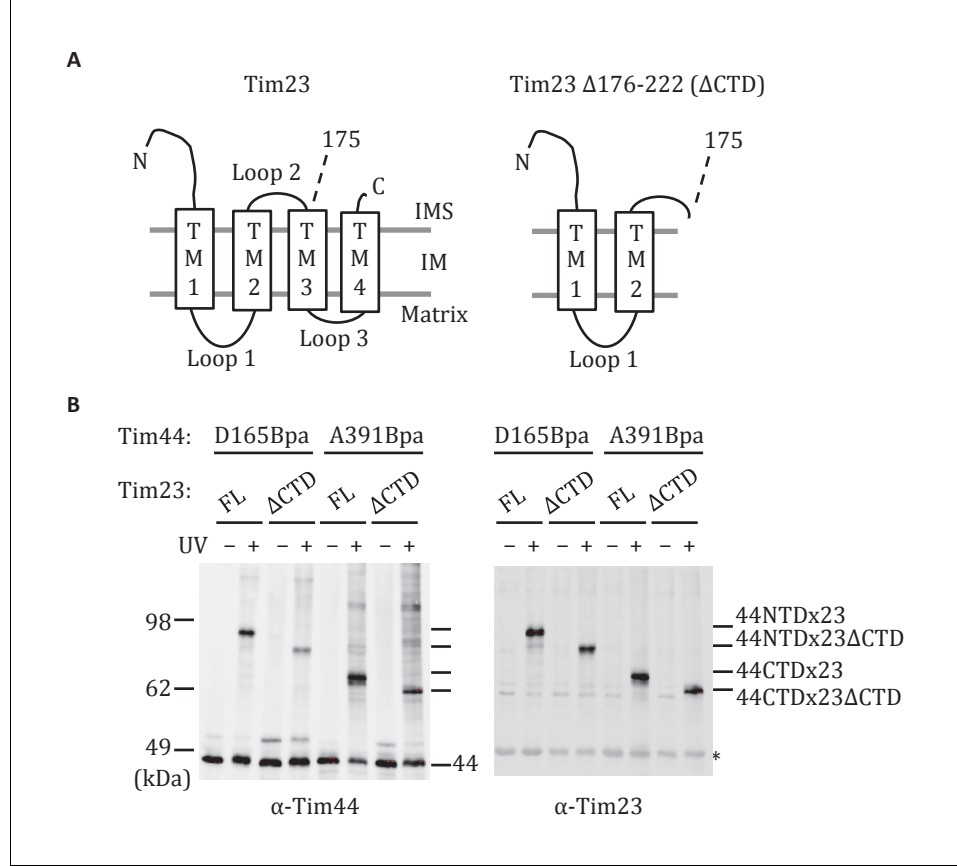

**Figure 4.** Both domains of Tim44 crosslink to loop 1 of Tim23. (**A**) Diagram of Tim23 (left) and Tim23$_{\Delta176-222}$ (right). Loops, transmembrane domains (TM), and residue 175 are indicated. (**B**) Yeast strains expressing Tim23 or Tim23$_{\Delta176-222}$ and Tim44 with Bpa incorporated at position 165 in the NTD or 391 in the CTD were subjected to UV irradiation (+), or as a control not exposed (-). Tim44 was then affinity purified via its N-terminal His$_6$ tag, followed by SDS-PAGE and immunoblotting with the indicated antibodies. Migration of size standards, in kDa, are indicted (left); Tim44 (44NTD or 44CTD) and Tim23 (23 or 23ΔCTD) reactive bands are indicated. The asterisk indicates non-specific cross-reactive bands.

$_{222}$) are viable (*Pareek et al., 2013*). We reasoned that if both domains of Tim44 interact with loop 1 of Tim23, Bpa variants in either domain would likely crosslink to Tim23 in cells expressing only *tim23$_{\Delta176-222}$*. However, if one domain interacted with loop 3, then no crosslink with that domain would occur. To carry out this test, we constructed strains expressing *tim23$_{\Delta176-222}$* and a variant Tim44 containing Bpa in either the NTD (Tim44$_{D165Bpa}$), which was previously found to crosslink to wt Tim23 (*Ting et al., 2014*), or CTD (Tim44$_{A391Bpa}$). In both cases, we observed a cross-reactive Tim44 crosslinking band after UV treatment (*Figure 4B*), which, as expected, migrated faster than those obtained when full length Tim23 was used. In both cases, the band also reacted with Tim23 antibodies, indicating that both domains of Tim44 interact with loop 1 of Tim23 in vivo.

## A Tim44 NTD alteration suppresses D345K CTD defects

As both domains of Tim44 can be crosslinked to loop 1 of Tim23, we carried out a suppressor analysis asking if a defect in translocon interaction in one domain could be compensated by an alteration in the other domain. We used the NTD R180K substitution, which destabilizes Tim44 association with the translocon (*Schiller et al., 2008*), and the CTD D345K substitution, described above, as starting mutations (*Figure 5A*). After constructing libraries using error-prone PCR, transformants were screened for the ability to grow at the non-permissive temperature of 34°C. No suppressors were found in the case of R180K. However, a suppressor encoding the single amino acid substitution G173V, which allowed *tim44$_{D345K}$* cells to grow as well as wt cells at 34°C, was isolated (*Figure 5B*). We used co-immunoprecipitation to test the effect of the G173V substitution on association with the translocon. Association of Tim44 was partially restored in the double mutant; co-immunoprecipitation was 74% of wt in *tim44$_{G173V/D345K}$*, rather than 45% found with *tim44$_{D345K}$* extracts (*Figure 5C*). We next asked if the G173V change also suppressed the defect in import caused by the D345K substitution (*Figure 1F*). Consistent with the robust growth of *tim44$_{G173V/D345K}$* at 30°C, *tim44$_{G173V/D345K}$* mitochondria imported cytochrome b$_2$-(167)$_{\Delta19}$-DHFR nearly as efficiently as wt mitochondria (*Figure 5D*). These results suggest that alteration of the NTD of Tim44 can partially overcome the effects of compromised association of Tim44 with the translocon due to alteration of the CTD.

To obtain information as to how the G173V suppressing substitution acts, we carried out Bpa crosslinking. We started with the double alanine CTD substitution mutation, *tim44$_{D345A/E350A}$*, whose 37°C growth defect is suppressed by the G173 substitution (*Figure 5E*). We started with this strain because its less severe phenotype allowed growth under conditions needed for Bpa incorporation. We constructed six strains. Each had the E350A/D345A mutations; three had the G173V suppressor substitution and three did not. Three different ±G173V suppressor pairs were then constructed, each having a stop codon for incorporation of Bpa at a different site: D165 of the Tim44 NTD and A391 of the Tim44 CTD, both of which crosslinks to Tim23; and S271, which crosslinks to Tim17 (*Figure 5F*). In all cases, we observed a cross-reactive Tim44 crosslinking band after UV treatment. The most obvious difference observed was an increase in crosslinking of D165Bpa to Tim23 when the suppressor mutation is present, suggesting that the G173V substitution alters the Tim44 NTD-Tim23 interaction.

## Interaction of the Tim44 NTD with presequence

Previously, full-length Tim44 was shown to interact in vitro with the presequence of Hsp60, implicating Tim44 in signaling the entry of a translocating polypeptide into the matrix (*Marom et al., 2011b*). In the previous report, biotin-labeled Hsp60 presequence peptide and a nonspecific crosslinker were used. We confirmed this result using the same approach, and then extended the analysis to the isolated NTD (Tim44$_{43-209}$) and CTD (Tim44$_{210-431}$). Interaction of the Hsp60 presequence with the NTD, but not the CTD was detected (*Figure 6A,B*). This binding was concentration-dependent (*Figure 6C*). To further narrow down the site(s) of presequence interaction, we attempted to purify NTD truncations. N-terminal truncations were unstable. However, we were successful in obtaining the C-terminal truncation segments containing residues 43–177, 43–170 and 43–164. While Tim44$_{43-177}$ crosslinked to presequence similarly to the complete N-terminal domain (Tim44$_{43-209}$), crosslinking between peptide and Tim44$_{43-164}$ or Tim44$_{43-170}$ were greatly reduced (*Figure 6D*). We also constructed three internal deletions in the NTD (Tim44$_{NTD\Delta68-82}$, Tim44$_{NTD\Delta131-147}$, Tim44$_{NTD\Delta166-185}$) to test whether these regions are important for presequence binding. While Tim44$_{NTD\Delta68-82}$, and Tim44$_{NTD\Delta131-147}$ crosslinked to presequence similarly to the intact NTD, crosslinking of presequence

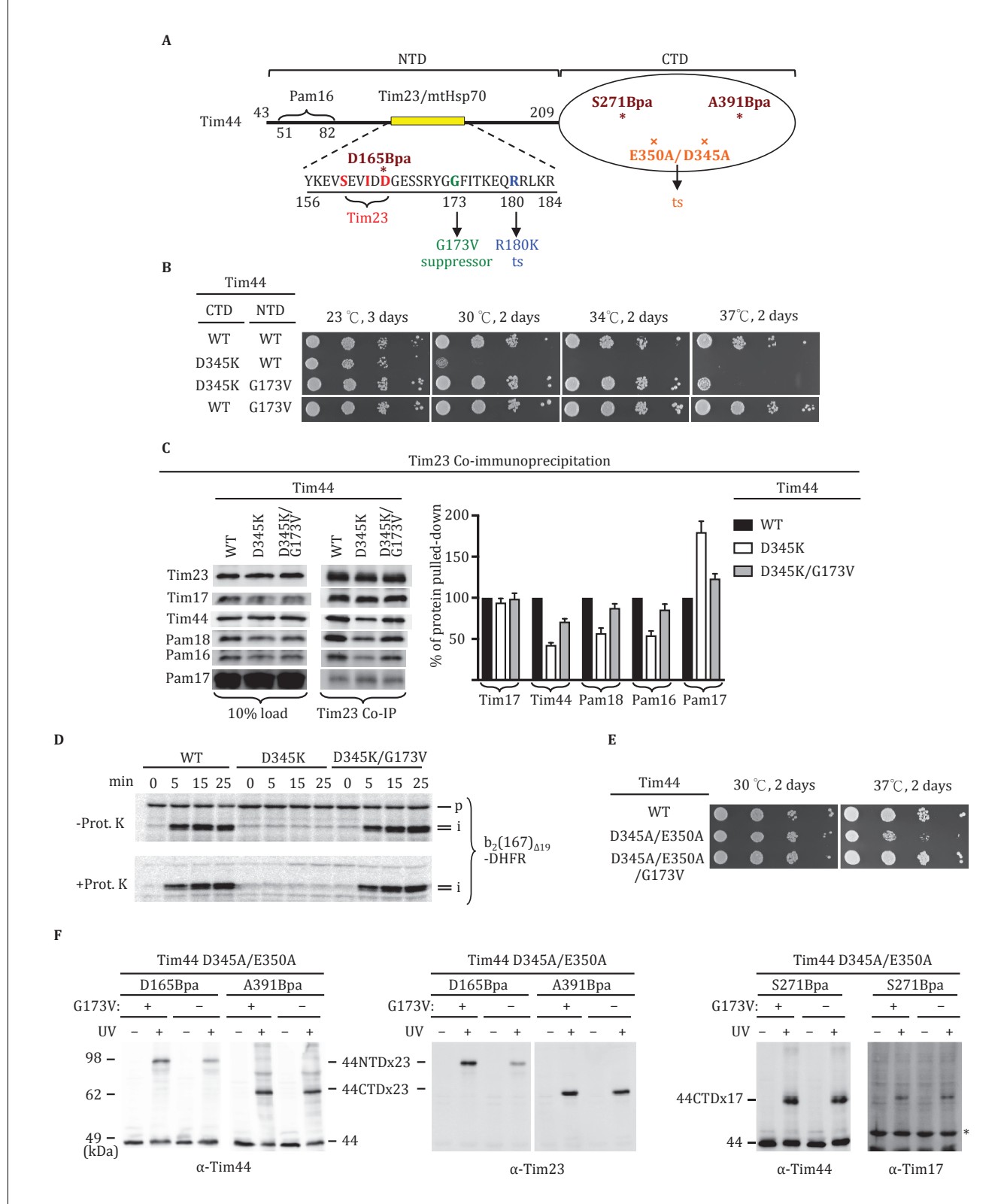

**Figure 5.** Tim44 NTD alteration suppresses D345K CTD defects. (**A**) Diagram of Tim44 showing functional regions and interaction partners of the NTD. Residues from 156 to 184 are indicated in yellow. Residues crosslinked to Tim23 in our previous report (*Ting et al., 2014*) are in red; residue G173, the suppressor reported here, and R180, the previously isolated ts mutant (*Schiller et al., 2008*), which affects Tim23 and mtHsp70 association, are in green and blue, respectively. The N terminal segment of Tim44 (residues 51 to 82) implicated in interaction with Pam16 is indicated (*Schilke et al., 2012*). The

*Figure 5 continued on next page*

*Figure 5 continued*

mutant D345A/E350A and residues D165Bpa, S271Bpa, A391Bpa used for crosslinking in combination the G173V suppressing substitution are indicated in orange and dark red, respectively. (B) Growth phenotype. Ten-fold serial dilutions of *tim44-Δ* cells expressing Tim44 variants were plated on minimal media and incubated at the indicated temperatures and times. For each temperature, all strains were plated on the same plate; strains not relevant to this report were cropped out (indicated by white space). (C) Co-immunoprecipitation with Tim23. Mitochondria were solubilized by treatment with digitonin. Solubilized material was subjected to immunoprecipitation using Tim23-specific antibodies crosslinked to protein A beads. Precipitates were analyzed by SDS-PAGE and immunoblotting using antibodies specific for the indicated proteins. 10% of input for co-immunoprecipitation was used as a loading control. Signals were quantified by Image J software (RRID: SCR_003070) and plotted as percentages of the wt control reaction value. Data represent the mean ± standard deviation, with n = 3. The data of the figure can also be seen in *Figure 5—source data 1*. (D) $^{35}$S-labeled precursor of cytochrome b2-(167)$_{\Delta 19}$-DHFR was imported into wild-type (WT), *tim44$_{D345AK}$*, and *tim44$_{D345AK/G173V}$* mitochondria at 25°C. Where indicated, mitochondria were subsequently treated with proteinase K (Prot. K). Samples were analyzed by SDS-PAGE and autoradiography. p, precursor; I, intermediate; m, mature. (E) Growth phenotype. Ten-fold serial dilutions of *tim44-Δ* cells expressing the indicated Tim44 variants were plated on minimal media and incubated at the temperatures and for the times indicated. (F) Yeast strains expressing *tim44$_{D345A/E350A}$* with Bpa incorporated at position D165, S271 or A391 were subjected to UV irradiation (+), or as a control not exposed (-). Tim44 was then affinity purified via its N-terminal His$_6$ tag, followed by SDS-PAGE and immunoblotting with the indicated antibodies. Migration of size standards, in kDa, are indicated (left); Tim44 (44NTD or 44CTD), Tim23 (23), and Tim17 (17) reactive bands are indicated. The asterisk indicates non-specific cross-reactive bands.

The following source data is available for figure 5:

**Source data 1.** Data analysis/quantification of Tim23 co-immunoprecipitation.

to Tim44$_{NTDΔ166-185}$ was reduced (*Figure 6E*). These results point to the particular importance of residues between positions 170 and 177 for Tim44's interaction with presequence.

## Evidence that the Tim44 NTD is intrinsically disordered

The finding that presequence binds to the Tim44 NTD, coupled with previous data pointing to its interaction with Tim23, mtHsp70 and Pam16, points to functional complexity of this domain. Since lack of structural information hampers understanding of the NTD, we attempted to obtain crystals of Tim44$_{43-209}$ for X-ray analysis. We were unsuccessful. We considered whether the NTD might be disordered, as it was well known to be very susceptible to proteolysis (*Weiss et al., 1999*). Indeed, the disorder prediction softwares, IUPred (*Dosztányi et al., 2005*), Metadisorder (*Kozlowski and Bujnicki, 2012*), and PrDOS (*Ishida and Kinoshita, 2007*), identified the NTD as having a high probability of being intrinsically disordered (*Figure 7A*). We decided to test this idea experimentally, by performing Heteronuclear Single Quantum Coherence (HSQC) and Circular Dichroism (CD) spectroscopy. The very limited chemical shift dispersion observed in the HSQC $^1$H dimension indicated that the segment is primarily disordered. In addition, the presence of different peak intensities in the HSQC is consistent with the idea that some regions of the NTD may dynamically exchange between multiple conformational states in the solution (*Figure 7B*). That the CD spectrum showed a minimum signal close to 200 nm, yet some signal at 222 nm, also suggests that the NTD is disordered and that it has some fractional transient helical propensity (*Figure 7C*).

## Discussion

Results presented here point to a three-pronged interaction of Tim44 with the translocon: two with Tim23 and one with Tim17 (*Figure 8A*). Overall our results point to the predominant role of one domain being tethering of Tim44 to the translocon, with the other serving as a dynamic arm, coordinating interactions amongst mtHsp70, J-protein co-chaperone Pam18 and the presequence of the incoming polypeptide.

### Interaction of CTD of Tim44 with the translocon

Adjacent patches on one face of the Tim44 CTD crosslinked to the translocon: one to Tim17 and one to Tim23. The phenotypes of mutants encoding alterations on this face suggest that both interactions play a role in stabilizing overall interaction of Tim44 with the translocon. Such mutations resulted not only in accumulation of precursors, but also the destabilization of the interaction with the translocon. The more dramatic effect of mutations changing residues in the Tim23-interaction region suggests that this interaction may be more functionally important. Recently, interaction of the

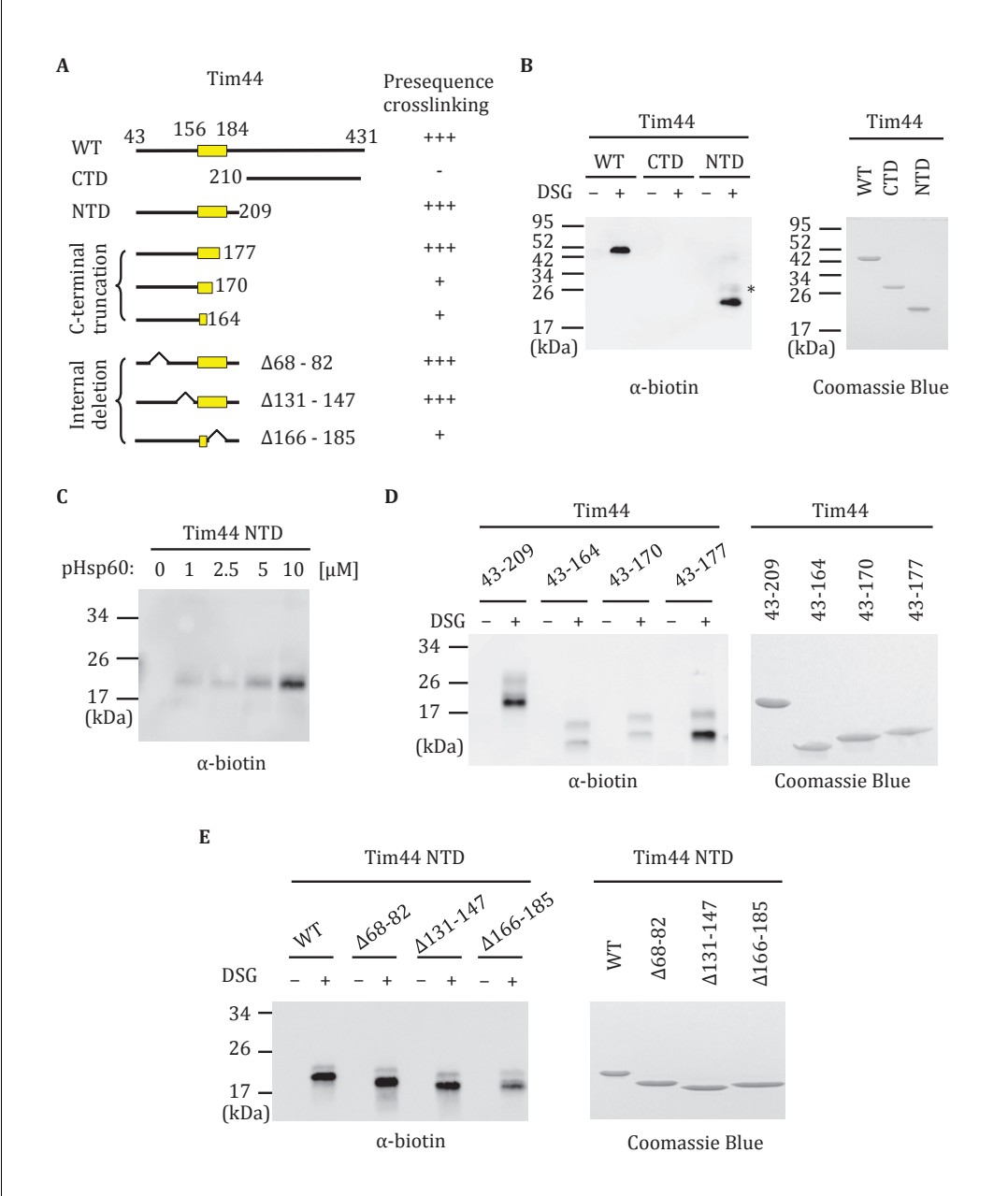

**Figure 6.** Interaction of the NTD of Tim44 with presequence. (**A**) Diagram of Tim44 showing different constructs along with a summary of crosslinking efficiency to the presequence of Hsp60 (pHsp60). Endpoint residue numbers are indicated. (**B,D,E** left) Crosslinking was performed by incubating biotin-labeled pHsp60 with indicated Tim44 proteins in the absence (-) or presence (+) of disuccinimidyl glutarate (DSG), followed by SDS-PAGE and immunoblotting using antibodies specific for biotin (streptavidin-HRP). (**B,D,E** right) Equal molar amounts of indicated Tim44 proteins were separated by SDS-PAGE and stained with Coomassie brilliant blue. (**B**) WT, full length Tim44 (residues 43–431); NTD (residues 43–209); CTD (residues 210–431). (**C**) Crosslinking of Tim44$_{43-209}$ with indicated increasing concentration of pHsp60. The asterisk indicates higher crosslinking products of the Tim44$_{43-209}$-pHsp60 complex. (**D**) Crosslinking of NTD and indicated C-terminal truncations. (**E**) Crosslinking of NTD and indicated internal deletions.

CTD of Tim44 with Tim17 was reported (*Banerjee et al., 2015*). However, interaction with Tim23 was not observed. Perhaps, the Tim44 CTD-Tim23 interaction has a more stringent conformational requirement than the Tim44 CTD-Tim17 interaction, as assays were performed in lysates of mitochondria disrupted by detergent. The crosslinking approach used here was performed using intact cells, and thus allows assessment of protein-protein interactions under native conditions. While this

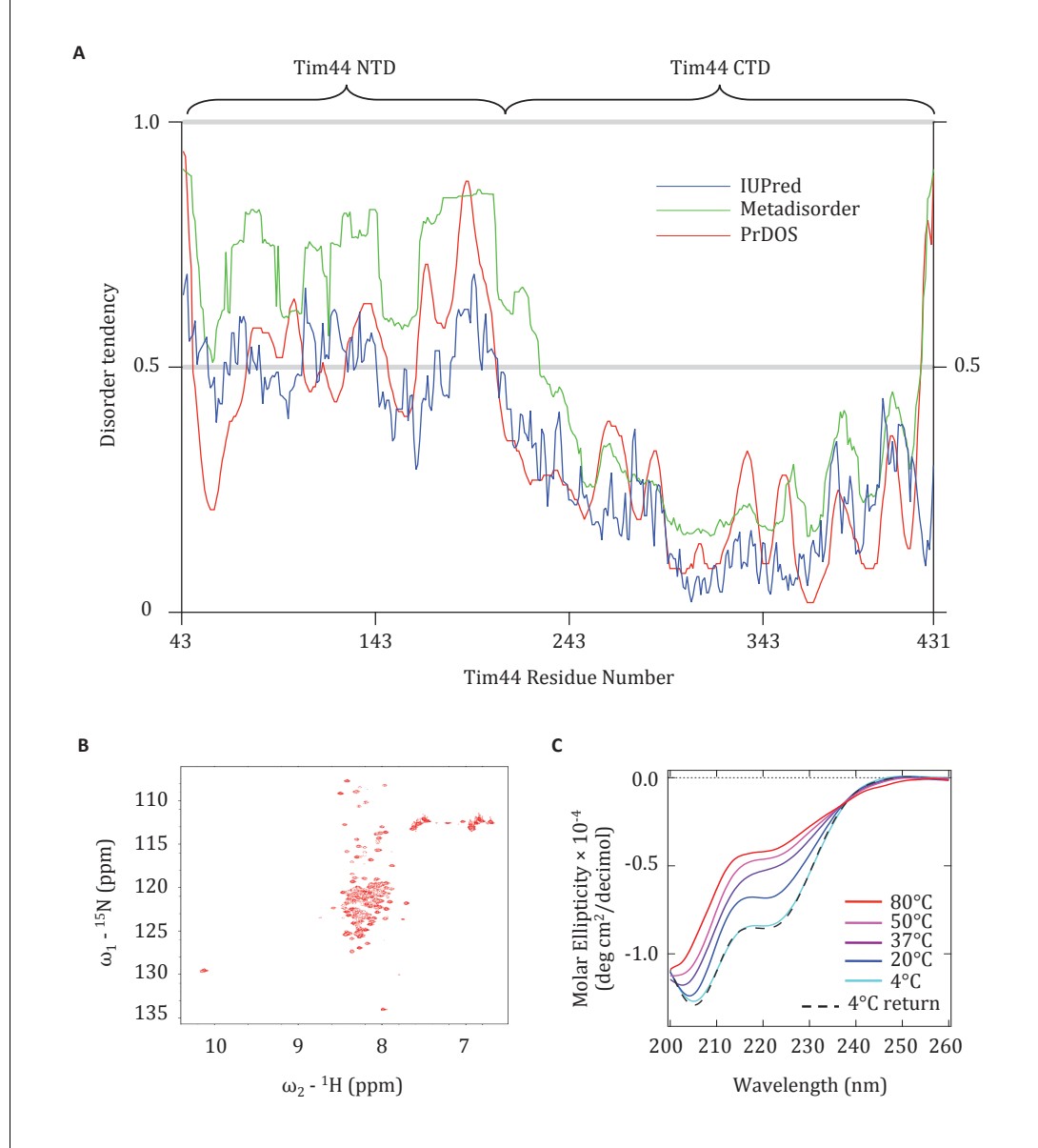

**Figure 7.** Evidence that the NTD of Tim44 is intrinsically disordered. (**A**) Prediction of disorder by indicated disorder predictors. The NTD and CTD of Tim44 are indicated at the top of the graph. Residues with a disorder tendency exceeding 0.5 were considered to be disordered. The data of the figure can also be seen in *Figure 7—source data 1*. (**B**) 2D $^1$H,$^{15}$N-Heteronuclear Single-Quantum Correlation (HSQC) NMR spectra of Tim44$_{43-209}$. As described in the text, the low dispersion of $^1$H$^N$ chemical shifts is indicative of an unfolded protein in solution. (**C**) Far-UV CD spectra of Tim44$_{43-209}$ as a function of temperature, recorded over the temperature range from 4°C to 80°C. After measuring the sample at 80°C, it was cooled down to 4°C to test the reversibility (4°C return). Smoothing spline model was used to fit the raw CD data. The data of the figure can also be seen in *Figure 7—source data 2*.

The following source data is available for figure 7:

**Source data 1.** Prediction of disorder by disorder prediction programs.

**Source data 2.** Circular Dichroism (CD) analysis of Tim44 NTD.

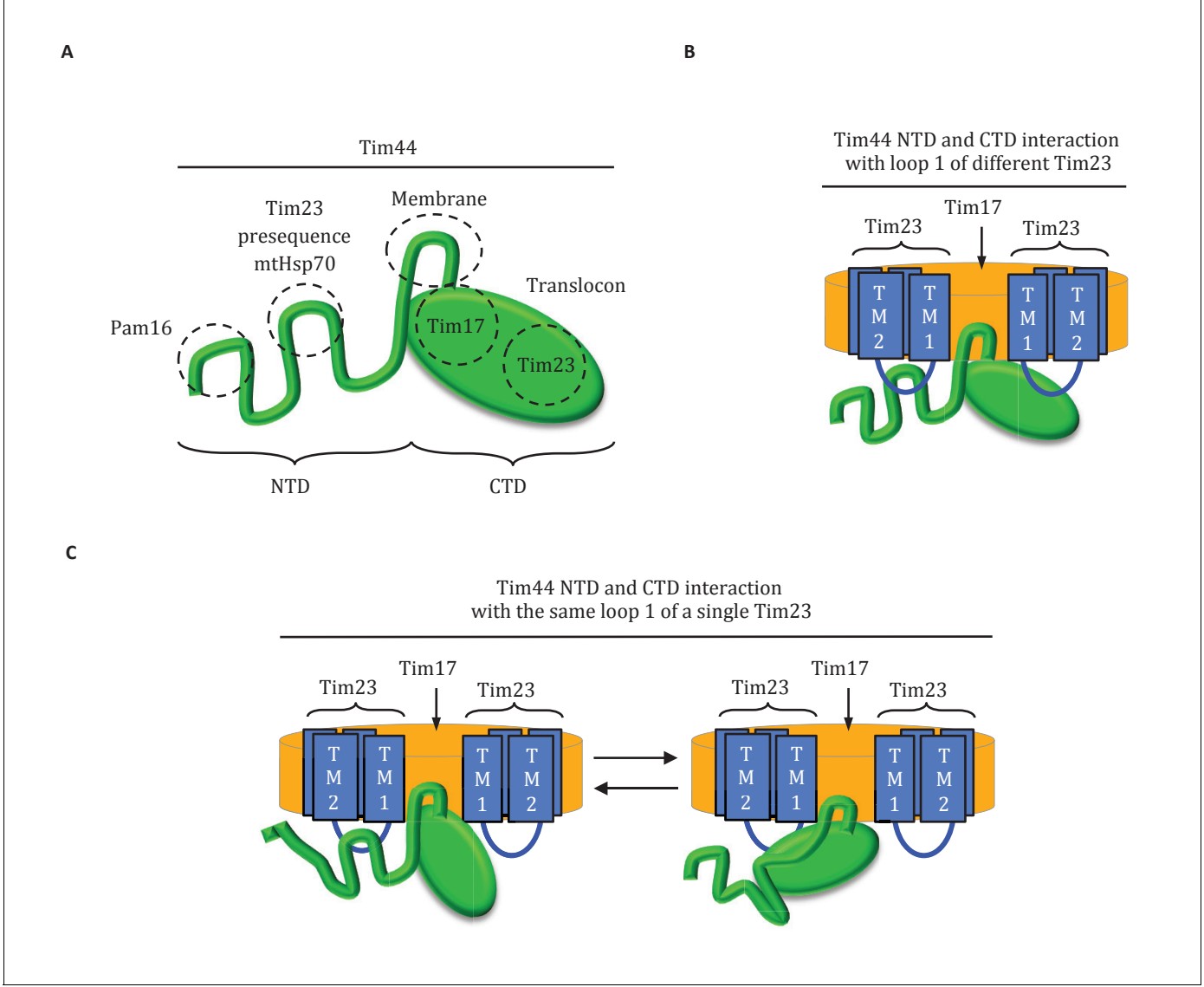

**Figure 8.** Diagrams of Tim44 interaction with the translocon. (**A**) Diagram of Tim44 (green) with the regions known to be important for interaction with other motor or translocon components indicated by dotted circles. The NTD is drawn to indicate its naturally disordered state. (**B**) Model of Tim44 interaction with loop 1 of two different Tim23 molecules (blue). Transmembrane domains 1 and 2 of Tim23 are indicated. Tim17 is indicated in orange. (**C**) Model of Tim44 NTD and CTD interacting with loop 1 of one Tim23 molecule, either the NTD (left) or CTD (right).

manuscript was under review, a report was published indicating the interaction of Tim44 with Tim17 occurs via loop 3 (*Demishtein-Zohary et al., 2017*), the smaller of the two matrix-exposed loops.

## Complexity of Tim23-Tim44 interactions

The complexity of the interaction of Tim44 with Tim23 is evident from the results presented here. Not only do both domains crosslink to Tim23, our data strongly indicates that both interact with loop 1; a C-terminal truncation variant of Tim23 lacking transmembrane domains 3 and 4, and thus loop 3, crosslinked to Tim44 whether Tim44 had Bpa in the NTD or in the CTD. That alteration of residues at or near positions that crosslinked to Tim23 affects the stability of Tim44's interaction with the translocon, suggests that both interaction sites are functionally important. This idea is also supported by the ability of the G173V alteration in the NTD to suppress the deleterious effects of a mutation altering the Tim23 interaction site in the CTD. However, the exact interplay amongst these

interactions remains elusive. In part, this lack of clarity is due to the fact that the overall architecture of the translocon is not known (*Schulz et al., 2015*) - neither the copy number of individual subunits, nor the exact positioning of their transmembrane domains relative to one another. There is consensus, however, that a translocon contains at least two molecules of Tim23 (*Alder et al., 2008*; *Marom et al., 2011a*). Thus, one Tim44 molecule could interact with loop 1 of two different Tim23 molecules simultaneously (*Figure 8B*). Alternatively, both the NTD and CTD of a Tim44 molecule could interact with a single Tim23 molecule (*Figure 8C*). If so, the interactions would likely occur sequentially, as the 24-residue size of loop 1 makes simultaneous interaction unlikely due to steric constraints.

## The multifunctional NTD of Tim44

Data presented here cements the idea that the NTD of Tim44 is a platform for protein-protein interactions, and further points to the region between residues 160 and 180 as a critical portion of the NTD. This segment was previously shown to be important for association of Tim44 with the translocon, and for recruiting mtHsp70 (*Schiller et al., 2008*). Indeed, the apparent increase of interaction of the NTD with Tim23 in the suppressor strain having a substitution at position G173 underscores this idea. Here, we provide evidence that it also includes the presequence binding site, suggesting that this segment not only has tethering functions, but also serves as a regulatory center for import motor function. It should also be noted that Pam16, which forms a heterodimer with the J-protein cochaperone of mtHsp70, binds Tim44 at the extreme N-terminus of the NTD (*Schilke et al., 2012*). That Tim44's NTD is likely intrinsically disordered is also in line with the idea that this domain plays important regulatory functions. Intrinsic disorder is a common feature of hub proteins that serve as regulatory scaffolds. Often domains of such proteins not only gain structure when interacting with partners, but also undergo conformational changes upon interacting with different binding partners (*Dyson and Wright, 2005*; *Oldfield and Dunker, 2014*; *Wright and Dyson, 2015*).

## Model of Tim44 action

How might the two domains of Tim44 work to coordinate activity of the import motor? We propose that Tim44 acts as a two-arm machine. One arm, the CTD, plays the major role in anchoring Tim44 to the translocase through interactions with both Tim17 and Tim23. The other, the NTD, is a dynamic arm. Although it also interacts with Tim23, this interaction plays a more regulatory role, with the NTD also interacting with Hsp70 and with Pam16, and thus indirectly with the J-protein co-chaperone Pam18, as well as presequence. Together the two domains control the inactive:active transition of the import motor. Key features of this working model include (*Figure 9*): (1) In the absence of a translocating polypeptide, the conformation of the Tim44 NTD maintains the import motor in an inactive state. Upon entering the matrix, presequence binds the Tim44 NTD in a region also important for interaction of the NTD with both Tim23 and Hsp70. (2) Presequence association initiates a conformational change in the Tim44 NTD reorienting Pam18 relative to Hsp70 such that it can perform its co-chaperone function of stimulating Hsp70's ATPase activity. (3) Hydrophobic residues of the presequence or adjacent regions of the preprotein interact, as a substrate, with Hsp70's peptide binding cleft. This interaction, coupled with that of Pam18, synergistically stimulates hydrolysis of ATP bound to Hsp70, stabilizing its interaction with the translocating polypeptide. (4) The resulting conformational change of Hsp70 (i.e. from the ATP- to the ADP-state) also results in release from Tim44 (*Liu et al., 2003*). This release of Hsp70 from Tim44 permits binding of another ATP-bound Hsp70 molecule primed to interact with the incoming polypeptide chain, and thus further translocation into the matrix.

Although the results presented here enlighten several aspects of the process of protein import into the mitochondrial matrix, several remain unresolved. For example, as mentioned above, whether the NTD and CTD of a Tim44 molecule interact with one or two Tim23 molecules is not clear. Interaction with two Tim23 molecules is the 'simplest' scenario; however, the alternative, that is toggling back and forth of the NTD and CTD between loop 1 of a single Tim23 molecule is appealing. Such toggling could provide an additional point of regulation or allow a major conformational change in the NTD. Apropos of this, it was recently reported that the Tim44 CTD could be crosslinked to a protein trapped in the translocon (*Banerjee et al., 2015*). The nature of such an interaction is not clear, as a nonspecific crosslinker was used and a substantial segment of the

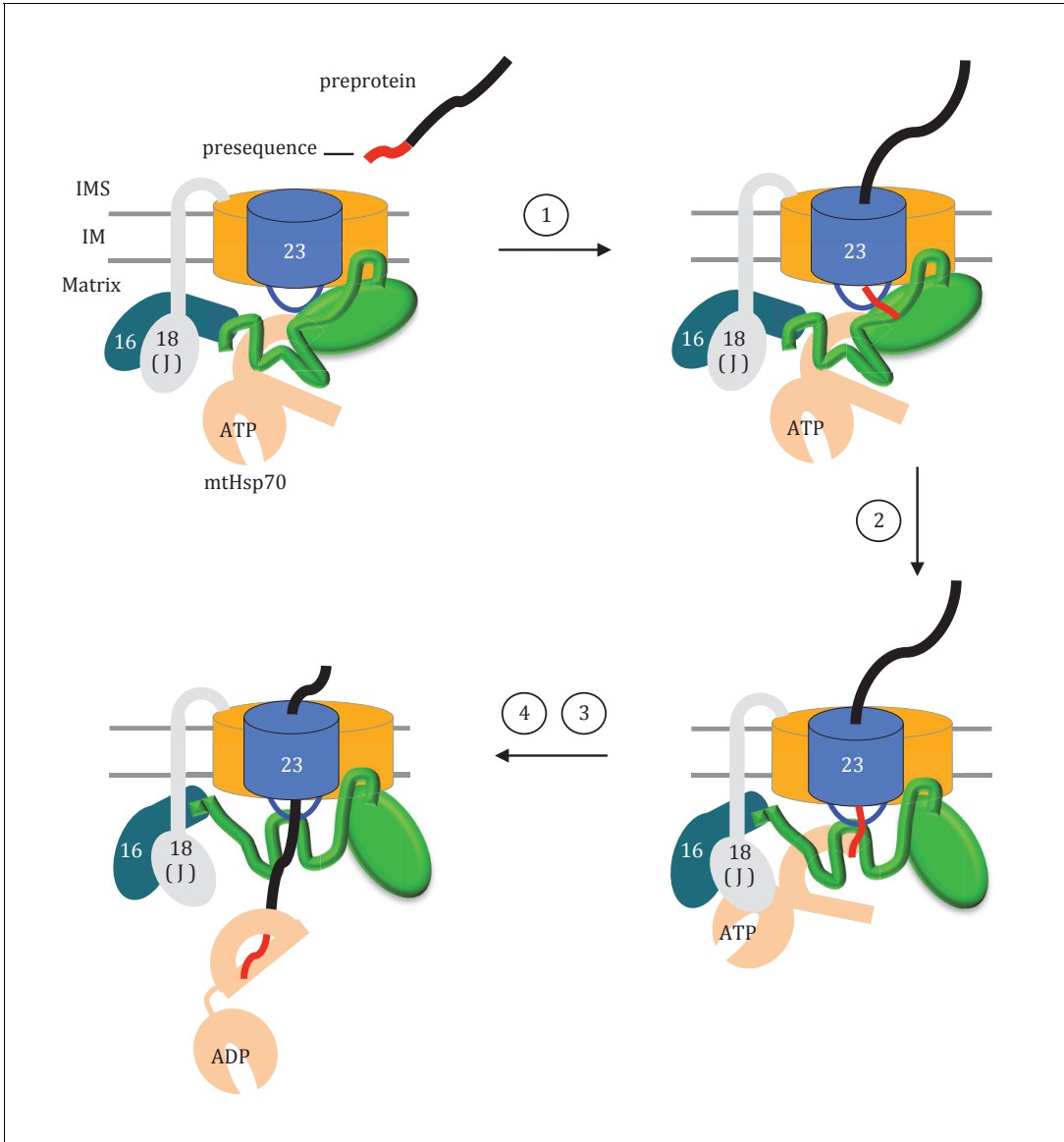

**Figure 9.** Model of Tim44 action during protein translocation. As in **Figure 8C**, one Tim44 is drawn as interacting with one Tim23 molecule. Pam16 interacts with the extreme N-terminus of the Tim44 NTD, and also interacts with the J-protein co-chaperone Pam18 ('J' indicates its J domain). Steps 1–4 are discussed in the discussion section of the text. Tim44 (green), with CTD (oval) and NTD (line); Tim 23 (blue); Tim17 (orange); Presequence (red line); remainder of translocating polypeptide chain (black line).

trapped polypeptide was present in the matrix. Nevertheless, such an interaction may help in maintaining the import motor in an active state during the entire translocation process. Also, it is possible that such toggling might control interactions of motor components, such as the proposed 'refreshing' of Pam18, at the translocon during the import process (**Schulz and Rehling, 2014**). Indeed, many aspects of the complex import system have dual and overlapping mechanisms that ensure the efficiency and robustness of the critical biological process of protein import into the mitochondrial matrix.

# Materials and methods

## Yeast strains, plasmids, and genetic techniques

All in vivo experiments were carried out in the W303 genetic background with derivatives of PJ53 (*James et al., 1997*). *tim44-Δ*, *tim23-Δ*, and *tim17-Δ* haploids were described previously (*Schilke et al., 2012*; *Schiller et al., 2008*; *Ting et al., 2014*). To create double deletion strains, the single deletion haploids were mated and the resulting diploids were transformed with the desired plasmids, sporulated and dissected. Screening for the correct deletion and plasmid marker genes identified the desired haploids. *tim44* point mutations and *tim23$_{Δ176-222}$* made in this study were constructed using the QuikChange protocol (Stratagene), starting with pRS314-*TIM44* or pRS315-*TIM23* as templates. pRS315-*tim23$_{Δ176-222}$* was generated by removing the *TIM23* DNA sequence from +526 to+666. Plasmid systems were set up for the incorporation of Bpa into Tim44 and Tim17 as previously described (*Krishnamurthy et al., 2011*; *Ting et al., 2014*). For Tim44 fragment purifications, the DNA sequence coding for various Tim44's N terminal segments were cloned into bacterial expression vector pET-20b with 5' NdeI and 3' BamHI sites.

To obtain temperature sensitive (ts) mutations in *TIM44*, that altered a surfaced exposed CTD residue, plasmid libraries were generated that contained DNA randomly mutagenized by error prone PCR amplification. More specifically, a PstI-BglII (+206 to +1253) fragment was amplified and used to replace the same fragment in pRS314-*TIM44*. *tim44Δ* cells carrying pRS316-*TIM44* (*URA3*-based plasmid) were transformed with the pRS314-*tim44* library, plated on tryptophan omission (Trp⁻) plates and incubated at 30°C. 1600 transformants obtained were patched onto 5-fluoroorotic acid (5-FOA) (Toronto Research Chemicals Inc., Toronto, Canada) plates and Trp⁻ plates and then incubated at 37°C to identify those cells that could not grow at 37°C in the absence of the wt copy of *TIM44*. The pRS314-*tim44* plasmids were recovered from patches on Trp⁻ plates, and then transformed into the parent strain to verify the ts phenotype. After selection for loss of the wt *TIM44* gene, the protein level of the Tim44 variants were determined by immunoblot analysis. 6 of 1600 transformants grew poorly at 37°C, yet expressed the Tim44 variant at a level similar to wt protein, and these plasmids were sequenced at the University of Wisconsin Biotechnology facility. Only one had alterations of surface-exposed CTD residues: glycine substitutions at residues D345 and R378. Single substitution analysis was carried out. R378G alteration caused no obvious phenotype, while D345G ts phenotype was indistinguishable from that of the original isolate.

A similar strategy was used to isolate suppressors of the ts phenotypes of the D345K and R180K substitutions. *tim44Δ* cells carrying pRS316-*TIM44* were transformed with pRS314-*tim44$_{D345K}$* or pRS314-*tim44$_{R180K}$* plasmids containing an N terminal (+206 to +876) or C terminal (+877 to +1253) mutation library. 1000 transformants of each were patched onto 5-FOA plates and incubated at 34°C. pRS314-*tim44* plasmids were recovered from transformants that grew at 34°C. No suppressors were found in the case of the R180K. However, 11 of the 1000 transformants allowed *tim44$_{D345K}$* cells to grow at 34°C. Among these, eight had a single substitution, G173V, and grew as well as wt at 34°C. Each of the other three suppressor strains, contained more than three substitutions in the NTD, and did not suppress as well as G173V.

## Mitochondrial protein precursor accumulation test

To test for the accumulation of precursors in vivo, *tim44* strains, except *tim44$_{D345A/E303A}$*, were grown in minimal medium at 30°C to an OD$_{600}$ of 0.5 at 30°C and then shifted to 37°C for 6 hr. Since *tim44$_{D345A/E303A}$* only grows at 23°C, cells were grown at that temperature prior to the shift to 37°C. Cell extracts were prepared from equal numbers of cells and analyzed by SDS-PAGE as previously described (*Schiller et al., 2008*). Immunoblotting was carried out using antibodies specific for the mitochondrial matrix protein Hsp60. Similar results were obtained from three independent yeast strains.

## Co-immunoprecipitation from mitochondrial lysates

As previously described (*Schilke et al., 2012*), association of Tim17, Tim44, Pam16, Pam18, and Pam17 with the Tim23 translocon was determined by co-immunoprecipitation. Antibodies against Tim23 were affinity purified prior to crosslinking to protein-A Sepharose beads (*D'Silva et al., 2005*). Mitochondria were isolated from Tim44 wt and mutant yeast strains grown at 30°C in rich

media, as described previously (*Liu et al., 2001*), with the exception of *tim44*$_{E303A/D345A}$ which was grown at 23°C. Mitochondria were solubilized at 1 mg/ml in lysis buffer (25 mM Tris-HCl, pH = 7.5, 10% glycerol, 80 mM KCl, 5 mM EDTA, and 1 mM PMSF) containing 1% digitonin (Acros Organics) on ice for 40 min with gentle mixing (*Mokranjac et al., 2003*). After spinning at 20,800 x g at 4°C for 15 min, the lysates were added to 25 µl (bed volume) of Tim23 antibody beads and incubated for 1.5 hr with mixing at 4°C. The beads were washed with lysis buffer without digitonin before boiling in SDS sample buffer. The proteins were separated by SDS-PAGE and detected by immunoblotting. Signals were quantified by Image J software (RRID: SCR_003070) and plotted as percentages of the wt control reaction value. Data represent the mean ± standard deviation, obtained in a minimum of three independent experiments.

## In organellar import of $^{35}$S-labeled preproteins

The import assay was carried out as described previously (*Hutu et al., 2008*). $^{35}$S-labeled preproteins, cytochrome b$_2$-(167)$_{\Delta 19}$-DHFR and cytochrome c$_1$, were synthesized by in vitro transcription and translation using the TNT SP6 Quick Coupled Transcription/Translation System (Promega) in the presence of [$^{35}$S] methionine (PerkinElmer). Preproteins were subsequently imported into isolated mitochondria in import buffer (250 mM sucrose, 80 mM KCl, 3% [w/v] bovine serum albumin, 5 mM MgCl$_2$, 5 mM methionine, 10 mM MOPS [morpholinepropanesulfonic acid]-KOH, pH = 7.2, 2 mM ATP and 2 mM NADH) for different time course intervals. The import reaction was stopped using 1 µM valinomycin, 8 µM antimycin, and 20 µM oligomycin. Protease treatment was performed by incubating the mitochondria with 50 µg/ml proteinase K (Prot. K) for 15 min. Prot. K was subsequently inhibited by the addition of 1 mM phenylmethylsulfonyl fluoride (PMSF). After Prot. K treatment, mitochondria were isolated by centrifugation and analyzed by SDS-PAGE, followed by soaking in gel fixing solution (10% glacial acetic acid, 25% isopropanol) for 30 min. The fixed gel was placed on a sheet of Whatman 3 MM filter paper, covered with plastic wrap and dried at 80°C for 1 hr under a vacuum using a conventional gel dryer (Bio-Rad). The gel was then exposed to a phosphor-imaging screen at room temperature overnight, and detected by a Typhoon FLA 9000 biomolecular imager (GE Healthcare).

## In vivo photo-crosslinking

Bpa crosslinking was carried out as described previously (*Ting et al., 2014*), with the plasmid ptRNA-Bpa, which encodes an engineered tRNA synthetase and tRNA$_{CUA}$ for Bpa incorporation. An amber stop codon (TAG) was introduced into the open reading frames of *TIM44 or TIM17* in the plasmids pRS314-*His$_6$-TIM44* or pRS414-*TEF-His$_6$-TIM17*, respectively. Cells were grown in the presence of 2 mM Bpa in minimal medium overnight. 50 OD$_{600}$ yeast cells were harvested, resuspended, and divided in two. One half was exposed to 365 nm UV (Stratalinker 1800 UV irradiator) for 1 hr at 4°C; the other half was kept on ice, as a control for no crosslinking. Cells were lysed via agitation with 0.5 mm glass beads in whole cell lysis buffer (20 mM HEPES-KOH, pH = 8.0, 10% glycerol, 500 mM NaCl, 20 mM imidazole, 0.2% Triton X-100, 1 mM DTT and 1 mM PMSF). Cell lysates were centrifuged for 20 min at 20,000 x g. Covalent protein adducts were isolated from the lysate via Ni-NTA purification system (Invitrogen), and analyzed by immunoblotting. Three independent yeast strains were analyzed for each Bpa variant, with similar results.

### Antibodies

Immunoblot analysis was carried out using standard techniques with protein detection by enhanced chemiluminescence (GE Healthcare), using polyclonal antibodies to Tim23 (*D'Silva et al., 2008*), Tim44 (*Liu et al., 2001*), Pam16 (*D'Silva et al., 2005*), Pam18 (*D'Silva et al., 2003*), Pam17 (*Schilke et al., 2012*), Hsp60 (*Schilke et al., 2012*), and Tim17 (gift from Nikolaus Pfanner) (*Chacinska et al., 2005*).

### Protein purification

Purification of Tim44, full length and CTD, were carried out as previously described (*Schiller et al., 2008*). For the NTD, the truncations and internal deletions, protein was expressed in *E. coli* Rosetta (DE3) at 37°C. Bacteria were grown in rich media to the 0.6 OD$_{600}$. Following 5.5 hr induction with 1.0 mM Isopropyl β-D-1-thiogalactopyranoside (IPTG), harvested cell pellets were resuspended in

lysis buffer (20 mM HEPES/KOH, 20 mM NaCl, pH = 7.5), followed by sonication, and then centrifugation for 20 min at 15, 000 x g. The obtained supernatant was loaded onto a cation exchange (SP Sepharose) column. After washing with buffer (20 mM HEPES/KOH, 30 mM NaCl, pH = 7.5), the protein fractions were eluted with a NaCl gradient (30 to 600 mM). The collected fractions containing Tim44 were further purified by gel filtration on a Superdex 75 column (GE Healthcare). The purified proteins were stored at −80°C in 20 mM HEPES/KOH, 300 mM NaCl, pH = 7.5.

### In vitro chemical crosslinking

Crosslinking of various Tim44 fragments was carried out as previously described (*Marom et al., 2011b*). In short, crosslinking was performed by incubating Tim44 fragments and biotin-labeled Hsp60 presequence (NH$_2$-MLRSSVVRARATLRPLLRRK-biotin) with 50 μM disuccinimidyl glutarate (DSG) at 25°C for 30 min. The reaction was stopped by the addition of SDS-containing sample buffer, followed by 65°C incubation. The crosslinking adducts were analyzed by immunoblotting with streptavidin-HRP (RRID: AB_1087562) (Thermo Scientific). Similar crosslinking results were obtained from three independent experiments.

## Heteronuclear Single Quantum Coherence and Circular Dichroism spectroscopy

The 2D $^1$H,$^{15}$N Heteronuclear Single-Quantum Correlation spectra were recorded on a 7.6 mg/ml (~380 μM) protein sample at the National Magnetic Resonance Facility at Madison (RRID: SCR_001449) (NMRFAM; www.nmrfam.wisc.edu) using a Varian VNMRS spectrometer operating at 800 MHz and equipped with a cryogenic probe. The temperature of the sample was regulated at 25°C throughout the experiments. NMR data were processed using NMRPipe and analyzed using NMRFAM-Sparky (*Lee et al., 2015*). All NMR experiments were conducted in 20 mM sodium phosphate (pH 7.5), 15 mM NaCl, and 7% D$_2$O. Similar spectra were obtained from three independent protein preparations.

CD data were collected using an AVIV model 420 spectrometer at the UW Biophysics instrumentation facility (BIF; www.biochem.wisc.edu/bif). Samples were 0.18 mg/mL (~9 μm) in 20 mM sodium phosphate buffer, pH = 7.5, 15 mM NaCl. Spectra were recorded with a 1 nm band-pass and an averaging time of 5 s in 0.1 cm path-length quartz cuvette. Data were collected every 1 nm from 340 to 200 and at five different temperatures: 4°C, 20°C, 37°C, 50°C, 80°C, then back to 4°C. Spectra were corrected by subtraction of a buffer baseline and converted into molar ellipticity using a mean residue weight of 118 (degree.cm$^2$/dmol of residues). Smoothing spline model was used to fit the raw CD data (*Wood and Jennings, 1979*). Similar spectra were obtained from two independent protein preparations.

## Miscellaneous

All chemicals, unless noted otherwise, came from Sigma. Hsp60 presequence peptide was purchased from the University of Wisconsin Biotechnology Center. Protein sequence alignment for Tim44 was carried out using Clustal Omega (RRID: SCR_001591) (*Sievers et al., 2011*). Intrinsically disordered protein prediction for Tim44 was carried out using IUPred (RRID: SCR_014632) (*Dosztányi et al., 2005*), MetaDisorder (*Kozlowski and Bujnicki, 2012*), PrDOS (*Ishida and Kinoshita, 2007*). Structural images were prepared with the PyMOL Molecular Graphics System (RRID: SCR_000305), Version 1.5 Schrödinger, LLC.

## Acknowledgements

We thank Jaroslaw Marszalek, Silvia Cavagnero and Katherine Henzler-Wildman for helpful discussions, Nikolaus Pfanner for Tim17-specific antibodies, and Marco Tonelli and Darrell McCaslin for guidance and use of software for processing structural data, as well as discussions. NMR data in the study made use of the National Magnetic Resonance Facility at Madison, which is supported by NIH grant P41GM103399 (NIGMS), old number: P41RR002301. Equipment was purchased with funds from the University of Wisconsin-Madison, the NIH P41GM103399, S10RR02781, S10RR08438, S10RR023438, S10RR025062, S10RR029220), the NSF (DMB-8415048, OIA-9977486, BIR-9214394), and the USDA. CD data were obtained at the Biophysics Instrumentation Facility (University of

Wisconsin – Madison) which was established with support from the University of Wisconsin - Madison and grants BIR-9512577 (NSF) and S10 RR13790 (NIH).

## Additional information

### Funding

| Funder | Grant reference number | Author |
| --- | --- | --- |
| National Institutes of Health | GM27870 | See-Yeun Ting<br>Nicholas L Yan<br>Brenda A Schilke<br>Elizabeth A Craig |
| Department of Biochemistry, University of Wisconsin-Madison | Steenbock Predoctoral Fellowship | See-Yeun Ting |

The funders had no role in study design, data collection and interpretation, or the decision to submit the work for publication.

### Author contributions

S-YT, substantial contributions to conception and design, analysis and interpretation of data, drafting the article and revising manuscript critically for important intellectual content; NLY, substantial contributions to analysis of data; BAS, acquisition, analysis and interpretation of data; EAC, substantial contributions to conception and design, analysis and interpretation of data, drafting the article and revising manuscript critically for important intellectual content, all authors approved of the version to be published

### Author ORCIDs

Elizabeth A Craig, http://orcid.org/0000-0002-9381-4307

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
