## [Decision Letter]

Thank you for submitting your article "Dual interaction of scaffold protein Tim44 of mitochondrial import motor with channel-forming translocase subunit Tim23" for consideration by *eLife*. Your article has been favorably evaluated by Randy Schekman (Senior Editor) and three reviewers, one of whom, Agnieszka Chacinska (Reviewer #1), is a member of our Board of Reviewing Editors.

The reviewers have discussed the reviews with one another and the Reviewing Editor has drafted this decision to help you prepare a revised submission.

The import of presequence-carrying proteins into the mitochondrial inner membrane and matrix is a process essential for eukaryotic life that is mediated by the TIM23 complex and its associated import motor. The study focuses on dissection of the mitochondrial protein translocase TIM23 architecture and dynamics. The conserved Tim44 protein is known as a central adapter component required for the cooperation between the transmembrane channel formed by Tim23 and the driving-force-generating matrix Hsp70 of the motor. In this work, Ting et al. present a detailed analysis of the interaction surfaces between Tim44 and the membrane-embedded core of the translocase using site-specific cross-linkers incorporated into Tim23, Tim17 or Tim44 in vivo. This detailed surface mapping together with the identification of the proposed presequence binding site in Tim44 considerably improves our understanding of preprotein import into mitochondria. The most important observations are: i) the Tim44 C-terminal domain binds both core components of the translocase, Tim17 and Tim23, and ii) binding to Tim23 is also mediated by the N-terminal domain of Tim44, which is intrinsically disordered.

The reviewers perceived the study as well presented and elegant. Furthermore, they appreciated its importance to progress the field. However, they agreed that several questions remain open. Addressing experimentally the points below is necessary to further advance the understanding of the mechanisms of the Tim44-mediated precursor translocation.

1) The authors should provide data allowing conclusions concerning translocase dynamics upon substrate binding. Do the interactions/crosslinking of Tim44 with Tim23 or Tim17 change by the presence of presequence peptides? And/or does the interaction mapping of Tim44 change during the mtHsp70 reaction cycle? These questions can be addressed in the assay with chemically synthesized peptides or under different nucleotide triphosphate conditions. An additional experimental set-up to consider: in vitro import of large amount of precursors into BPA containing mitochondria and crosslinking.

2) Figure 7; The purified recombinant NTD did not show any indication of folded structures in its NMR or CD spectra. However, this does not mean that NTD is unfolded in intact mitochondria. There are lots of examples that recombinant proteins expressed in *E. coli* cells cannot take correctly folded structures. Therefore the interpretation of Figure 7, " the Tim44 NTD is intrinsically disordered", should be more softened or substantiated by additional evidence. The model presented in this manuscript implies that Tim44 NTD undergoes substantial conformational rearrangements upon presequence binding. The authors should take the opportunity to test this idea in their established assay (Figure 7) using the pHsp60 peptide (Figure 6) to experimentally support their model.

3) Earlier studies have shown that alterations of Tim44 can influence the overall conformation of the TIM23 complex (for example recently in Banerjee et al. 2015, *eLife*). The Tim23 and Tim44 steady state protein levels and pHsp60 accumulation experiments shown in Figure 1 and Figure 2 are very general indicators. It is important to analyse critical mutants via in vitro import experiments, if the observed defects are caused by motor coupling to TIM23 or by a general defect of the import channel. This can be done by comparing the import behavior of well-established motor-dependent and -independent preproteins in vitro. Alternatively, the authors could confirm that the TIM23 core complex is not altered in the above mentioned mutants. The sentence in the Discussion tells that the interaction with Tim23 is functionally more important than Tim17, but it is unclear what is the data source for this statement. More thorough characteristic of the selected mutants in terms of import and translocase integrity should provide the answer.

4) The suppression of Tim44-C Tim23 binding by the mutation in the N-terminal domain should be further developed by demonstrating this phenomenon at the level of translocase integrity and in vitro import experiments. How does the suppressor mutation really work: by strengthening the binding of Tim44 to Tim23, or maybe to Tim17? Addressing this point would add to Tim44 dynamics and its possible mechanistic implications. Interestingly the suppressor mutation exactly maps to the region of presequence binding. The suppression may therefore also be caused by altered (enhanced?) presequence interaction of Tim44. It would be very interesting to also consider this possibility.

---

## [Author Response]

[…] The reviewers perceived the study as well presented and elegant. Furthermore, they appreciated its importance to progress the field. However, they agreed that several questions remain open. Addressing experimentally the points below is necessary to further advance the understanding of the mechanisms of the Tim44-mediated precursor translocation.

1) The authors should provide data allowing conclusions concerning translocase dynamics upon substrate binding. Do the interactions/crosslinking of Tim44 with Tim23 or Tim17 change by the presence of presequence peptides? And/or does the interaction mapping of Tim44 change during the mtHsp70 reaction cycle? These questions can be addressed in the assay with chemically synthesized peptides or under different nucleotide triphosphate conditions. An additional experimental set-up to consider: in vitro import of large amount of precursors into BPA containing mitochondria and crosslinking.

We have tried both approaches (mitochondria/crosslinking earlier and mitoplasts during revision) with Bpa incorporated at a few positions. We did not see dramatic effects. This could be for several reasons. Of course, our working model that a conformational change of Tim44 and/or its interacting partners occurs upon presequence binding could be wrong. But there are other possibilities, as well, for why we did not see changes. For example, in the case of the import assays, a low efficiency of translocon loading would make changes difficult to detect; and although we think we had substantial loading, accurate calculation of this is difficult. We think that this question requires a more systematic comprehensive approach, using many more Bpa variants than we have in hand at this point. It would be also important to test positions that show very little crosslinking (or even negative) in in vivo crosslinking assays, as well as those that are positive. This is a long-term project.

2) Figure 7; The purified recombinant NTD did not show any indication of folded structures in its NMR or CD spectra. However, this does not mean that NTD is unfolded in intact mitochondria. There are lots of examples that recombinant proteins expressed in E. coli cells cannot take correctly folded structures. Therefore the interpretation of Figure 7, " the Tim44 NTD is intrinsically disordered", should be more softened or substantiated by additional evidence. The model presented in this manuscript implies that Tim44 NTD undergoes substantial conformational rearrangements upon presequence binding. The authors should take the opportunity to test this idea in their established assay (Figure 7) using the pHsp60 peptide (Figure 6) to experimentally support their model.

We agree with the reviewer that a protein being intrinsically disordered does not mean that it does not attain structurein vivo; probably most intrinsically disordered proteins do when they are binding their partner proteins. We have added a phrase to the Discussion to try to better emphasize this point. In addition, we have changed the Abstract to clearly indicate that both the presequence binding and characterization of the NTD are the results of experiments done in vitro. Also, the title of the legend of Figure 7, which contains the NMR and CD analysis, has been changed from “The Tim44 NTD is intrinsically disordered…” to “Evidence that Tim44 NTD is intrinsically disordered…”, which matches the title of that section in the Results.

A couple of years ago we did obtain spectra of the NTD after addition of presequence peptide. We didn’t see a dramatic change. We actually weren’t surprised, as other interacting partners (i.e. Pam16, Hsp70 and Tim23) were not present. We did not continue this direction because of the difficulty in obtaining all the purified components, particularly in the large amounts needed for NMR experiments.

3) Earlier studies have shown that alterations of Tim44 can influence the overall conformation of the TIM23 complex (for example recently in Banerjee et al. 2015, eLife). The Tim23 and Tim44 steady state protein levels and pHsp60 accumulation experiments shown in Figure 1 and Figure 2 are very general indicators. It is important to analyse critical mutants via in vitro import experiments, if the observed defects are caused by motor coupling to TIM23 or by a general defect of the import channel. This can be done by comparing the import behavior of well-established motor-dependent and -independent preproteins in vitro. Alternatively, the authors could confirm that the TIM23 core complex is not altered in the above mentioned mutants. The sentence in the Discussion tells that the interaction with Tim23 is functionally more important than Tim17, but it is unclear what is the data source for this statement. More thorough characteristic of the selected mutants in terms of import and translocase integrity should provide the answer.

We have carried out *in organellar* import assays for two *TIM44* mutants, *tim44_D345K_* and *tim44_E303A/D345A_*, with two different precursors that have been used previously (e.g. analysis of *tim44-804* in Hutu et al. [2008] MBoC 19:2642): cytb_2_-(167)_∆19_-DHFR and cytochrome c_1_, destined for the matrix and inner membrane, respectively. Both mutants were very defective in import of cyb_2_-(167)_∆19_-DHFR, but not cytochrome c_1._ This data has been added to Figure 1 and Figure 2.

The sentence in the Discussion “The more dramatic effect of mutations changing residues in the Tim23-interaction region suggests that this interaction may be more functionally important” was based on the comparative growth defects of the double alanine substitutions, that is D345A in combination with each of the other 5 conserved charged residues, individually, shown in Figure 2. Of these, the D345A/E303A double mutant was by far the most compromised, barely forming colonies at 23 °C. D345A/E295A and D345A/E350A were compromised at 37 °C, but not 30 °C. Residue E303 is in close proximity to the Tim23 crosslinking patch shown in Figure 3 and Figure 3—figure supplement 1. In the case of the weaker genetic interactions – residues E350 and E295 are in close proximity to the Tim17 patch. We have also constructed charge reversal mutations for each of the 5 conserved charge residues. Only the E303K change showed a phenotype. It grew as poorly as D345K. This result was not shown in the original submission. We have now included as Figure 2.

*4) The suppression of Tim44-C Tim23 binding by the mutation in the N-terminal domain should be further developed by demonstrating this phenomenon at the level of translocase integrity and* in vitro *import experiments. How does the suppressor mutation really work: by strengthening the binding of Tim44 to Tim23, or maybe to Tim17? Addressing this point would add to Tim44 dynamics and its possible mechanistic implications. Interestingly the suppressor mutation exactly maps to the region of presequence binding. The suppression may therefore also be caused by altered (enhanced?) presequence interaction of Tim44. It would be very interesting to also consider this possibility.*

We have extended the analysis of the G173V suppressor in two ways. First we tested the effect of the suppressor on import of cyb_2_-(167)_∆19_-DHFR into mitochondria from cells having the double G173V/D345K substitution in Tim44 (D345K on its own causes a severe defect; see Figure 1). As is the case of the severe growth defect at 30 °C, the G173V substitution overcomes the severe *in organellar* import defect. This result is now shown in Figure 5.

We also carried out an additional experiment using Bpa crosslinking to see if we could detect a difference in crosslinking upon suppression. To do this we combined two CTD substitutions (D345A/E350A) that result in a 37 °C ts phenotype that is suppressed by G173V. We then created amber stop codons to allow Bpa incorporation at 3 sites (D165, crosslink site of Tim44 NTD with Tim23; A391, crosslink site of Tim44 CTD with Tim23; S271, crosslink site of Tim44 CTD with Tim17). We found that the Tim44 NTD-Tim23 crosslinking was higher in the presence of G173V, while the Tim44 CTD (i.e. Tim23 and Tim17) crosslinks were not significantly changed. This result is consistent with the suppressor mutation resulting in increased interaction with the Tim44 NTD. Although, much additional work will be required to firmly establish the mode of action of the suppressor. These results are shown in Figure 5.